# BioTEA: Containerized Methods of Analysis for Microarray-Based Transcriptomics Data

**DOI:** 10.3390/biology11091346

**Published:** 2022-09-13

**Authors:** Luca Visentin, Giorgia Scarpellino, Giorgia Chinigò, Luca Munaron, Federico Alessandro Ruffinatti

**Affiliations:** Department of Life Sciences and Systems Biology, University of Turin, 10123 Turin, Italy

**Keywords:** transcriptomics, microarray, differential expression, reproducibility

## Abstract

**Simple Summary:**

Researchers are often interested in detecting whether there are any differences in gene expression levels between two types of cells. To do this, gene expression levels are measured, and specific computer programs are used to detect these differences. Historically, microarrays were used to measure gene expression, but they are now being supplanted by newer, more efficient technologies such as RNA sequencing. BioTEA allows users to perform the differential expression analysis of microarray-derived data easily, quickly and in a reproducible way. It combines all the steps needed to directly start from the gene expression levels and obtain a list of genes that are differentially expressed between the two cell types of interest. In this way, the large amount of publicly available microarray data can still be analyzed in the modern era. Differential expression analyses can be rather complex to run, but BioTEA makes them straightforward, so that even non-bioinformaticians can perform them with ease. BioTEA is free and open-source.

**Abstract:**

Tens of thousands of gene expression data sets describing a variety of model organisms in many different pathophysiological conditions are currently stored in publicly available databases such as the Gene Expression Omnibus (GEO) and ArrayExpress (AE). As microarray technology is giving way to RNA-seq, it becomes strategic to develop high-level tools of analysis to preserve access to this huge amount of information through the most sophisticated methods of data preparation and processing developed over the years, while ensuring, at the same time, the reproducibility of the results. To meet this need, here we present bioTEA (biological Transcript Expression Analyzer), a novel software tool that combines ease of use with the versatility and power of an R/Bioconductor-based differential expression analysis, starting from raw data retrieval and preparation to gene annotation. BioTEA is an R-coded pipeline, wrapped in a Python-based command line interface and containerized with Docker technology. The user can choose among multiple options—including gene filtering, batch effect handling, sample pairing, statistical test type—to adapt the algorithm flow to the structure of the particular data set. All these options are saved in a single text file, which can be easily shared between different laboratories to deterministically reproduce the results. In addition, a detailed log file provides accurate information about each step of the analysis. Overall, these features make bioTEA an invaluable tool for both bioinformaticians and wet-lab biologists interested in transcriptomics. BioTEA is free and open-source.

## 1. Introduction

Gene expression DNA microarrays [1] were introduced in the mid-1990s and represented the first cost-effective *-omics* technology for transcriptome profiling [2]. Such a technology involves the hybridization of fluorescently labeled cDNA with solid microchips exposing a collection of thousands of known short DNA sequences attached to the chip surface in a defined position and serving as probes. Commercial and custom-designed microarrays evolved through two main generations of chips: the first *two-color* (or *two-channel*) spotted microarrays [3] and the next *single-channel* (or *one-color*) high-density microarrays [4], mainly popularized by Affymetrix (Santa Clara, California) and capable of a much more reliable genome-wide quantification of the expression changes across multiple samples and different experimental conditions. The need for acquiring, processing and analyzing the ensuing high-throughput transcriptomics data streams has been a powerful driver for the development of innovative computational methods and statistics approaches within the context of bioinformatics across the postgenomic era [5,6,7,8,9]. The birth and the rapid growth of the Bioconductor project [10], together with the two most important public repositories for gene expression data—namely the Gene Expression Omnibus (GEO) [11,12] and ArrayExpress (AE) [13,14]—is probably what best embodies the deep impact of that first technological revolution in *-omics* disciplines.

However, starting from the late 2000s, the nascent Next-Generation Sequencing (NGS) technology fostered a second paradigm shift in transcriptomics, leading to the development of the RNA-seq technique [15], the current standard for transcriptome quantification. RNA-seq is increasingly taking hold because, by sequencing, aligning and counting cDNA fragment “reads”, it is possible to overcome many of the limitations inherent to microarrays, among which are the necessary dependence on known genome sequences, a relatively high background signal (leading in turn to a poor limit of detection), a narrow (2–3 orders of magnitude) dynamic range due to the saturation of the fluorescent spots and the challenging normalization procedures needed to compare expression levels among different experiments.

As a consequence, starting from 2015, scientific publications referring to expression microarrays have shown a downward trend, having been overtaken around the same year by an increasing number of RNA-seq-related papers, which, to date, are still in a phase of exponential growth [16]. These publishing trends clearly show that, because of its higher accuracy and thanks to the constant lowering of NGS prices, RNA-seq is likely to completely replace microarray technology and, as a result, all the related methods of data analysis are meant to change accordingly. In particular, while many high-level algorithms developed to analyze microarray data (i.e., starting from the normalized expression matrix on) have already been effectively ported or adapted to RNA-seq normalized counts, most of the low-level preprocessing, quality control and normalization techniques are specific to microarrays, if not even platform-specific (Affymetrix, Agilent or Illumina). From such a perspective, the risk of an imminent and general loss of the technical know-how needed to analyze expression data from microarray experiments is real.

Nevertheless, we believe that there are a number of good reasons to keep this expertise alive or, even better, to *freeze and package* our current gold standards of microarray data analysis in order to provide present and future researchers with suitable tools to deal with this kind of data at every occurrence:Despite the transcriptomics trends, thousands of articles referring to microarray experiments or microarray analyses are still published yearly [16], and this is likely to continue for a while;The aforementioned GEO and AE databases contain a huge amount of microarray raw data (more than $10^{4}$ studies overall) that still deserve to be explored by reanalyses (e.g., Sabaie et al. [17]) or meta-analyses (see e.g., Leal-Calvo and Moraes [18]);There is a growing concern about the reproducibility of research results in biomedical sciences and bioinformatics [19,20,21,22];Typical pipelines for microarray analysis are custom scripts made up of multiple files and several R functions from different Bioconductor packages; dealing with this code—and correctly running a new analysis—months or years later can be source of frustration for many researchers, even among bioinformaticians.

To meet all these needs and make microarray data analysis more accessible to the broad community of biologists, we developed bioTEA (biological Transcript Expression Analyzer), a stand-alone Dockerized pipeline for the retrieval, preprocessing, differential expression analysis and annotation of transcriptomic data from one-color (high-density) microarrays. Its containerized design ensures the reproducibility of results over time and across multiple informatics platforms, providing a friendly Command Line Interface (CLI) that frees the end user from coding, and R syntax in particular. Notably, although specifically designed with microarray technology in mind, bioTEA can handle RNA-seq data provided in the form of read counts, thus extending its scope of applicability and making it a powerful and invaluable bioinformatics tool.

## 2. Materials and Methods

BioTEA can be used to perform all steps of a classical Differential Expression Analysis (DEA): data retrieval (from GEO), data preprocessing, differential expression analysis by either empirical Bayes-moderated *t*-test (i.e., limma R package [23]) and/or Rank Product statistics [24,25] and result annotation with metadata. The user is presented with an intuitive, easy-to-use CLI to perform every step of the analysis, which is installed as biotea. A standard bioTEA analysis involves the following 5 steps:With biotea retrieve, data are directly downloaded from GEO, notably with sample metadata included that will be useful, although not required, to perform successive analysis steps.With biotea prepare, raw expression data (as returned by the microarray scanner setup, or as downloaded according to the previous step) are read, parsed, normalized and quality-controlled. The normalized expression matrix is saved to file, along with Quality Control (QC) plots.With biotea initialize, the differential expression analysis is initialized, optionally by parsing metadata about the samples. This allows the user to quickly set a variety of analysis options, such as variables of interest (e.g., treatment status), contrast of interest (e.g., “treated” vs. “control”) and analysis batches for batch effect correction. This step generates an options file that records the various choices made. This allows the user to inspect, edit or share them for reproducibility. Additional information regarding the parameters that can be set for the analysis are included in the Appendix A.With biotea analyze, the options file (such as from the previous step) and the expression matrix are read and DEA is performed. This generates several QC plots, along with differential expression tables, as a final result. Conservative filtering is also performed on the data before the analysis, to increase statistical power.With biotea annotations, annotations (such as gene symbol or gene name) can be added to expression matrix files or differential expression tables to allow further analyses (such as Gene Ontology (GO) enrichment analysis) and considerations.

It is important to note that these steps need not be run in order, as they are independent of each other. For example, if expression data are already available from other sources, one may directly run biotea analyze.

Reproducibility was a key goal in the design of bioTEA. Data preparation, analysis and annotation is performed by code hosted inside a Docker container. This allows for the computational reproducibility of these steps, while preserving the automated nature of the tool. Options for the core DEA steps are inputted through an options file, allowing investigators to share the analysis quickly and easily. Indeed, given the same input files and options file, the container assures identical analysis output, independent of operating system version or hardware. Finally, all bioTEA commands log their output in log files, allowing further inspection of run-time events.

### 2.1. The bioTEA Container

BioTEA is divided into two intimately related components: the bioTEA Docker container and the bioTEA CLI. The bioTEA container is structured in modules that perform different functions, all written in the R programming language. A single shared entrypoint binds all modules together. The container performs all the core analysis steps, namely data preparation, DEA and annotation. The general structure of the container and its modules is shown in Figure 1.

Calling the container directly is complex, as several arguments are needed both to configure the Docker daemon (e.g., input–output mounting paths) and run the modules themselves. To make this task more accessible to the end user, the bioTEA python package was created. This package exposes the biotea command, and handles argument parsing and the correct invocation of the Docker daemon. It also provides additional helper commands that are useful in the overall analysis. To promote reproducibility, no data analysis is performed by the bioTEA python package directly.

In the following sections, we aim to explain in more detail the bioTEA container modules, the key pipeline choices we made while writing the core analysis code and the structure and usage of the bioTEA python package.

### 2.2. Microarray Data Preparation

The two preparatory modules prepaffy and prepagil can be used to handle the preprocessing of Affymetrix and Agilent microarray raw data, respectively. As input, both modules take a series of text files generated by the specific microarray scanning apparatus. Then, the following steps are performed.

Find and load all input files. prepaffy searches the target folder for all files ending with the .CEL file extension, while prepagil searches by default all .txt files. However, as .txt is a very common file extension, prepagil search criteria can be customized in the call, also supporting Regular Expressions.Merge all inputs into a single expression data object. Both commands do this automatically using oligo and limma packages for Affymetrix and Agilent data, respectively. From this step onward, data are expressed as $\log_{2}$ values.Generate QC plots before normalization. For each sample, a Bland–Altman plot (MA plot) is saved, along with an overall expression boxplot.Normalize the expression data. Both data types are background-subtracted (through the *normexp* algorithm) and interarray-normalized (through the *quantile–quantile* procedure). Additionally, for Affymetrix data, the Robust Multichip Average (RMA) procedure as provided by the oligo package is used to collapse individual probes into the probe set to which they belong.Generate new QC plots after normalization. New MA plots and an overall expression boxplot are saved to appreciate the effects of normalization.If specified, remove control probes from the data set. In particular, negative control probes (i.e., sequences designed to remain unhybridized) represent a sensible estimate for the expected intensity of unexpressed genes as a result of background and nonspecific hybridization signals. Such an estimate is outputted and logged when handling Agilent data, as it can be used as a filter threshold value in the downstream analysis. This step is carried out by bioTEA for Agilent data, and by the oligo package for Affymetrix data.Collapse the replicate probes in Agilent arrays. Data are collapsed by taking the mean value of replicate probes found inside each sample.Save the final expression matrix as the output file. The expression matrix is saved in .csv format, ready to be analyzed by the other modules of bioTEA, or with customized pipelines.

For more information on the input–output data structures and formats, please refer to Appendix B.

### 2.3. Performing Differential Expression Analysis

The analysis module handles the core DEA, which consists in detecting those genes showing different levels of expression across two or more experimental groups, each made up of many biological replicate samples. To this end, samples are labeled using the levels of an input variable named “*experimental design*” (e.g., “treatment1”, “treatment2”, “control”). Special shorthand notation is available to label a large number of samples at once (see the Appendix A for more information). Differentially Expressed Genes (DEGs) are searched between these groups of samples, as indicated by the *contrasts* variable (e.g., “treatment1” vs. “control”). Any number of contrasts can be tested in a single run. Beyond a number of parameters—including the experimental design and the contrasts of interest—the module takes as input an expression matrix such as the one generated by the prep- modules.

The analysis can be divided into 5 steps: optional (re-)normalization, input visualization, filtering, statistical analysis, output visualization. First, data can be *quantile–quantile* normalized (which is useful only if they were not normalized beforehand). Diagnostic plots are saved before and after this step.

Samples are then clustered by means of a hierarchical algorithm and through Principal Component Analysis (PCA). A dendrogram is saved along with the distribution of the samples in the space of the first principal components. Both representations are useful to spot possible batch effects, which may have profound influence on the final DEG lists if not properly accounted for. In addition, a mean variance plot for each experimental condition is included for further QC inspections.

Expression data are then filtered to remove low-expression genes and increase the statistical power of the subsequent tests. Filtering is accomplished in a conservative way, according to three parameters: κ being the minimum group-wise presence, $\theta_{E}$ being a threshold value on $\log_{2}$ expression levels and $\theta_{F}$ being a threshold on $\log_{2}$ fold change values. An entry (a single probe or probe set, or a transcript cluster, depending on the input type) is retained only if its expression is above $\theta_{E}$ in at least κ percentage of samples of at least one experimental group. In addition, while not removed outright, genes that result in $\left|\log_{2}FC\right|<\theta_{F}$ will be marked as “non-differentially expressed” in the final output, even if they show statistical significance, because they usually have little biological importance and can be difficult to validate through other means (such as quantitative PCR). However, they are still considered in the computation.

The detection of DEGs is handled by limma [23], one of the de facto standards for this purpose, and by RankProd [24,25], a non-parametric alternative. Limma detects DEGs by fitting linear models and performing a gene-wise moderated *t*-test taking advantage of empirical Bayes methods to return more reliable *p*-values even in the case of small sample sizes. Notably, since limma linear models can take into account a variety of different variables, the user may define a “batch” variable to specify which batch the samples originate from. Batch effect correction is handled during the statistical analysis, both in the RankProd and limma tests. We chose not to correct the batch effect *a priori*, as recent literature suggested that this may lead to data distortion and wrong conclusions [26].

After running limma and/or rankprod, bioTEA generates a series of diagnostic plots, including MA plots and volcano plots. If both the limma and RankProd analyses are run, a comparison of the two outputs in the form of Venn diagrams is also included.

The output of the analysis module consists in a series of .csv format files, one per input contrast per analysis tool (either RankProd or limma or both), named “DEG tables”. Each table contains every retained entry from the input file, along with information about test statistics, *p*-values, False Discovery Rate (FDR) *q*-values, as well as average expression and $\log_{2}FC$. Moreover, an additional “markings” column is included, labeling each entry as significantly upregulated (1), downregulated (−1) or non-differentially expressed (0). An entry is labeled as differentially expressed if it features a *q*-value strictly less than the critical FDR/Proportion of False Positives (PFP) value of 0.05, and if its average $\log_{2}FC$ is greater than $\theta_{F}$ or less than $-\theta_{F}$, as noted before.

If specified in the input, the DEG tables can be automatically annotated and gene symbols are also used to enrich some output plots, such as volcano plots.

As an option, bioTEA can also analyze RNA-seq data in the form of counts, such as those generated from the use of HTSeq [27]. The discrete count data are transformed into continuous expression data using the voom function [28] provided by the limma package.

### 2.4. Annotating Results

The annotate module adds additional metadata to either expression matrices or DEG tables. The metadata can be sourced by an internal, static annotation file generated with 39 human microarray annotation packages present on Bioconductor. This was done for reproducibility reasons, so that the same annotations are applied while using the same bioTEA version. However, packaging all annotations present in Bioconductor, spanning multiple species and microarrays, would render the container too heavy memory-wise. In any case, bioTEA may be configured to download annotation data from Bioconductor at run time, filling this gap.

Added annotations include gene name and Human Genome Organization (HUGO) gene symbol, as well as which package was sourced to obtain the data, and its version.

### 2.5. The bioTEA Command Line Interface

The bioTEA CLI is an application written in Python 3 and compatible with all UNIX-like operating systems. It allows for easy access to the bioTEA container and for additional features, such as the automatic retrieval of data from GEO. As it is packaged and published on PyPA, installation is straightforward with the standard Python utility pip.

The bioTEA CLI handles the download, update and removal of the bioTEA containers automatically or on demand of the user.

### 2.6. Source Code Availability and Installation

The bioTEA command line interface is available as a Python package on PyPI [29], and its code, as well as the code used in the bioTEA container, is available on GitHub [30]. As already noted, a pre-compiled version of the bioTEA image is available at Docker Hub [31].

For reproducibility, bioTEA follows Semantic Versioning 2 for its versioning schema [32]. At the time of writing, the latest bioTEA CLI version is v1.0.1, and the current bioTEA Docker image version is v1.0.1. The bioTEA versions might change as patches and bug fixes are introduced in the future. It is important to note that the versions of the two components might diverge; however, we strive to keep compatibility between the two components with identical major and minor versions. The bioTEA CLI warns the user of possible compatibility issues automatically.

To install the tool, install Python version 3.9 or later, as well as the Docker engine and daemon. Afterwards, install bioTEA with pip install bioTEA. The pip utility handles the download, installation and update of the tool. Afterwards, the bioTEA CLI may be invoked with the biotea command.

## 3. Results

In this section, we exemplify the usage of bioTEA by analyzing a publicly available data set. This example analysis was run with bioTEA version 1.0.1. The Docker engine was pre-installed according to the official documentation (server version 20.10.14). BioTEA was installed with the pip install biotea command inside a Python 3.10 virtual environment. Please refer to the Appendix A for the exact sequence of commands used to run the analysis presented in this section.

### 3.1. Data Retrieval

We chose to re-analyze the data provided by Zhang et al. [33,34], regarding samples of pancreatic cancer matched with nearby healthy tissues. Data were retrieved directly from GEO (accession GSE28735) with biotea retrieve. This command downloads and generates a number of files in the target directory: a metadata.csv file and two sub-directories: a raw_data directory with the compressed samples and a unpacked_samples folder with the uncompressed samples.

### 3.2. Preprocessing and Quality Control

To preprocess the data, uncompressed samples were converted to flat .CEL files with gunzip. Then, we processed the raw samples to obtain the expression matrix with biotea prepare affymetrix. The result is a series of QC plots, saved in a new PrepAffy Figures folder, and an expression matrix file, saved in .csv format.

The QC plots are composed of two MA plots per sample, one before normalization and one after normalization. In each case, the *M* statistic is calculated between the sample and the median value for the same entry in all other samples. The so-generated plots are enumerated, starting from the sample that produced the less linear Loess fit, allowing the user to detect at a glance the samples with lower quality, especially in experiments with very large sample sizes. Two of these plots can be seen in Figure 2. Additionally, two global boxplots and density plots are generated, before and after data normalization, to appreciate the effects of the normalization steps.

### 3.3. Differential Expression Analysis

The options file for biotea analyze was generated with biotea initialize and then manually edited to set the appropriate parameters for the analysis. A copy of such an edited options file is included in the Appendix A. The analysis was then launched with biotea analyze. The result is a collection of files generated in the target output folder, which includes a series of QC plots, as well as the output DEG tables.

Samples are automatically renamed according to the levels of the main variable of interest, which, in this example, are either “tumor” or “normal”. Importantly, the order of the input samples matters, as the labels are applied to the samples order-wise, from left to right in the expression matrix. For this reason, a correspondence matrix showing the new labels along with the original sample names is saved in the output directory for manual correctness checks.

QC plots include a boxplot and a density plot of the input data, similar to those produced by biotea prepare, to check that all the samples are properly interarray normalized. A clustering dendrogram and a PCA plot are also included, to detect possible batch effects (Figure 3). MA plots comparing samples of the different groups of interest are saved. The same plots are additionally saved with DEGs—as detected by both limma and RankProduct—overlaid, as shown in Figure 4.

A very common representation of the results of a DEA is the volcano plot. BioTEA generates one volcano plot per tool per contrast (see Figure 5).

The large sample size of the data set (n=45 for each condition) and its exceptional quality allowed for high statistical power, reflected in the high number of detected DEGs. The detected DEGs were concordant between limma and RankProduct: both detected the same 70 downregulated genes, while RankProd detected 3 more upregulated DEGs (271) compared to limma (268). The shared upregulated genes were identical between the two tools.

## 4. Discussion

In this paper, we presented bioTEA as a new tool for the analysis of transcriptomics data. Thanks to the virtualization technology offered by Docker, our software aims at being an operating-system-independent and robust solution to obtain reproducible results in differential gene expression analysis. Because of its Python wrapper, bioTEA can be easy installed and run as a high-level command-line application, not requiring any specific programming skill by the user. The main idea behind this project was to use containerization to “hibernate” the most advanced and consolidated standards developed in the last several decades for the analysis of microarray data. We believe that this will allow researchers to preserve the practical ability of mining the huge amount of publicly available microarray data sets, even in the realistic context of the forthcoming decline of this technology.

Notably, throughout the software development process, bioTEA has been enriched with many capabilities and features not initially foreseen, including the possibility of analyzing RNA-seq data, thus greatly extending its scope of applicability. Importantly, although designed to be easy-to-use, bioTEA does not sacrifice analysis power and versatility, with the current version of bioTEA providing the following notable features:Functional reproducibility ensured by Docker container technology;R/Bioconductor-independent CLI through Python wrapper;Automatic data set retrieval and metadata parsing (from GEO);Microarray raw data processing capability;Microarray multi-platform support (Agilent and Affymetrix);Support for RNA-seq data analysis through the voom package [28];Automatic annotation for many human arrays (no chip ID needed);Low-expression, conservative gene filtering;Easy experimental design definition by syntax parsing;Two-independent-class or paired design testing;Double approach to DEA (parametric by limma and non-parametric by RankProduct);Batch effect handling;Additional explanatory variable support;Detailed logging;High-quality graphical output;Detailed wiki user guide;Open-source and easily accessible code;Modular code structure suitable for further development and maintainability.

Even if many user-friendly tools for gene expression data analysis have been proposed over the years, to our knowledge, none of them encompass all of bioTEA’s features listed above. For example, GEO2R [35] can only analyze GEO-available data sets, while BioTEA can handle data from any source. Additionally, it does not perform any preliminary filtering, it only supports a limma-based analysis and it does not support sample pairings, batch effect handling or complex limma models. It is also not highly reproducible, as it only uses the latest versions of R packages. BART (bioinformatics array research tool) [36] is conceptually very similar to BioTEA, but has a few key differences. First, it is used through a web application. This makes sharing the options used in a specific analysis more difficult than with BioTEA. Most importantly, BART is not containerized, and uses whatever R version is locally available. Even though BART’s authors recommend using their remote-hosted instance of BART, it is not available at the time of writing (August 2022). MeV (MultiExperiment Viewer) [37] is an extended tool for performing DEAs. However, MeV’s latest standalone release was in 2013, and it uses an obsolete R version (v2.11.1). Additionally, it is not Dockerized, does not handle batch effects and provides a graphical interface to the analysis, which has similar pitfalls to BART. Illumina’s BaseSpace provides a streamlined and powerful DeSeq2-based pipeline for performing DEAs starting from RNA-seq data. However, its usage is not free, and the analysis code is closed-source. Additionally, to the authors’ knowledge, it exclusively supports RNA-seq data.

## 5. Conclusions

We believe that BioTEA provides an invaluable tool to inexperienced and veteran bioinformaticians alike. For the inexperienced, the ease of use of the tool and its extensive documentation make data preparation and DEAs easy and reproducible. More experienced users will undoubtedly appreciate the simple nature of the output, and the possibility to run many analyses efficiently.

To reproduce a bioTEA analysis, it is sufficient to share the command used to prepare the data and the analysis options file. This renders the analysis process more transparent for the whole community. Additionally, the reproduction of the analysis can be performed even without extensive knowledge of any programming language.

The code for bioTEA, together with a detailed installation and usage guide, is available online and licensed under the permissive MIT license at https://github.com/CMA-Lab/bioTEA, accessed on 6 September 2022. The BioTEA containers are available at Docker Hub at https://hub.docker.com/u/cmalabscience (accessed on 6 September 2022) for manual inspection; BioTEA downloads the necessary containers automatically at run time. Users are invited to inspect BioTEA’s code, propose changes to it or open issues regarding it. The extensive documentation in the form of function doc-strings and comments throughout the code, as well as its modular nature, allow experienced programmers to easily read and modify the tool to suit their own particular needs.

## Figures and Tables

**Figure 1 biology-11-01346-f001:**
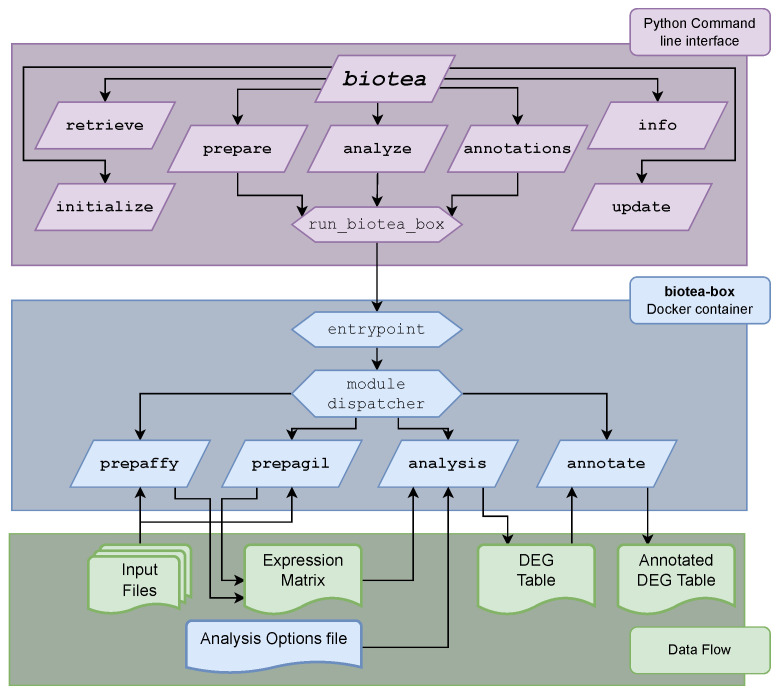
General structure of bioTEA. The python CLI is accessed with the biotea command. Launching of the Docker container is handled by a single function, run_biotea_box. A shared entrypoint is run upon starting the Docker container, which parses the options and runs the invoked module. File input and output are handled through Docker mounts to predetermined locations inside the running image. In the lower part of the figure, we provide a simplified schema of the input and outputs used by the various modules.

**Figure 2 biology-11-01346-f002:**
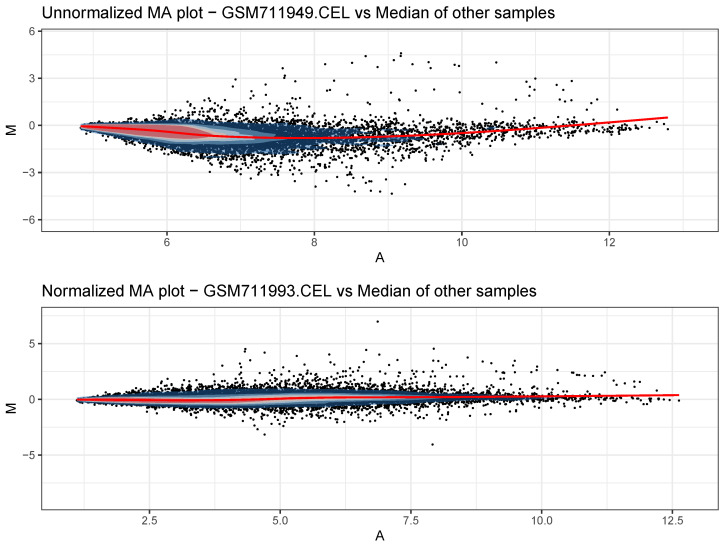
Two Bland–Altman plots showing the distributions of the genes before (**top**) and after (**bottom**) normalization procedures. Data taken from the Zhang data set.

**Figure 3 biology-11-01346-f003:**
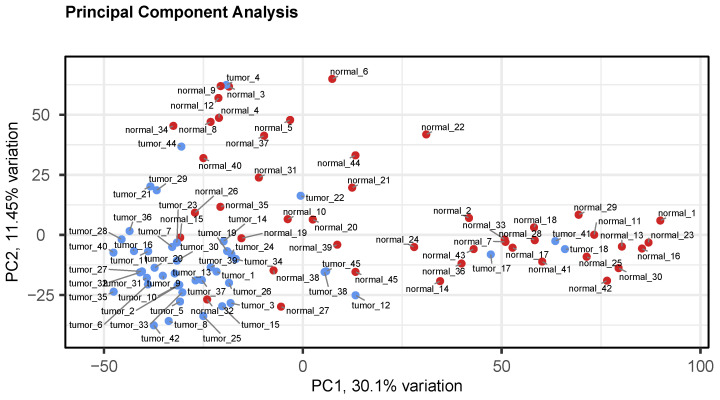
A PCA plot of each sample in the Zhang data set. The samples are well clustered according to their different sample status (tumor or normal) and no other effect seems to be present. Data taken from the Zhang data set.

**Figure 4 biology-11-01346-f004:**
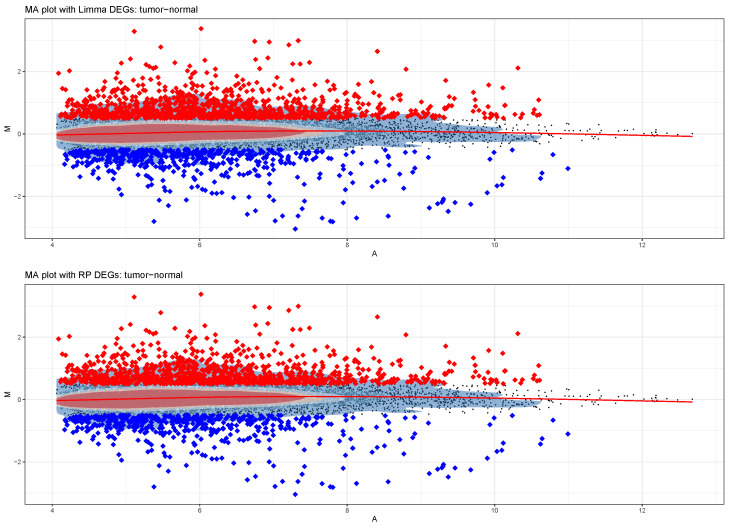
Bland–Altman plots showing the detected DEGs as either upregulated (red) or downregulated (blue), as detected by limma (**top**) or RankProduct (**bottom**) in the Zhang data set. The two plots look identical, a sign that DEG detection was robust.

**Figure 5 biology-11-01346-f005:**
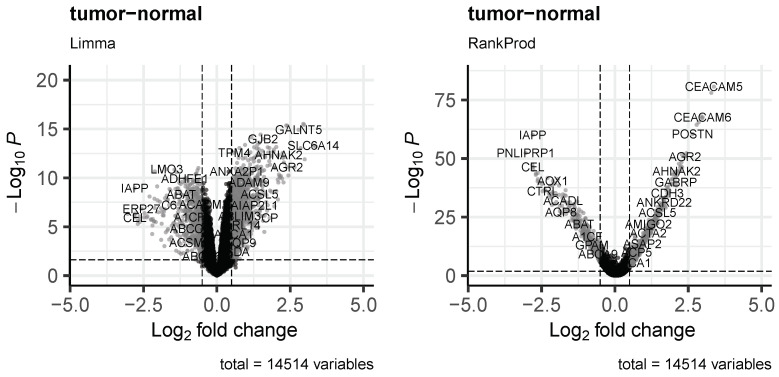
Volcano plots of the DEGs detected by limma (**left**) and RankProduct (**right**) in the Zhang data set. The different shapes of the distributions are related to how the two algorithms compute the *p*-values. The horizontal dashed line is the $-\log_{10}\left(p-\mathrm{value}\right)$ threshold corresponding to an FDR alpha level of 0.05. The vertical lines are the $\log_{2}$ fold-change thresholds, which can be changed by the user and are by default set to 0.5.

## Data Availability

Data used to exemplify the usage of the tool can be accessed on GEO with accession number GSE28735.

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
