# Peer review of "BioTEA: Containerized Methods of Analysis for Microarray-Based Transcriptomics Data"

_biology, 2022, doi:10.3390/biology11091346_

Round 1

Reviewer 1 Report

The authors introduce bioTEA as a novel tool for analyzing expression data from microarray experiments to keep the current gold standards of microarray data analysis alive in versus the highly growing technology; NGS. They illustrated several advantages, parts, and method of working of bioTEA.

In general, the introduced software is impressive, however the article is missing whether there is a similar softwares in the market ? what is advantages of the bioTEA software in comparison to such others ?

Besides, some language corrections such as:

Line 2: a many should be “many”

Line 3: Array Express (abbrev. should be added AE)  

Line 21: “consisted in” should be “involves”

Line 326: OS stand for operating system

Author Response

We thank Reviewer 1 for the high praise they gave to our manuscript. In regards to their concerns, we had already mentioned some tools that were similar in scope to BioTEA. However, we had not gone into detail regarding their specific differences. We have now done so, having expanded the relevant section in the Discussion. Moreover, we have added a mention of a commercially-available tool, Illumina's BaseSpace, and highlighted its differences with BioTEA.

We thank Reviewer 1 for the grammatical and spelling errors they pointed out. They have been fixed in the revised manuscript.

Reviewer 2 Report

In the manuscript “BioTEA: containerized methods of analysis for microarray-based transcriptomics data" by Luca Visentin et al., the authors present and propose a novel free robust methodology for transcriptomics big data analysis. The theme is interesting and extremely relevant for the field. The paper is well-written and organized. However, in the whole text, the authors fail to disclose the website through which the tool is available, even though it is stated that “The code for bioTEA is available online and licensed under the permissive MIT license. Users are invited to inspect the code, propose changes to it, or open issues regarding it.”  Therefore, it is not possible to evaluate and understand the full capabilities of the tool presented. Thus, it will be mandatory to correct this key aspect before the paper is considered suitable to be published in Biology.

Author Response

We thank Reviewer 2 for the praise given to our manuscript.

We apologise for the obscure inclusion of the link to BioTEA's code and usage guides. We had only included them in the bibilography. All the relevant links are now clearly included in the main text, both in the abstract and the conclusion sections.

Reviewer 3 Report

This manuscript is of interest; however, it has the features of a technical note; the authors lack information to evaluate if this work reaches the level for an artificial intelligence application.
I suggest restructuring it as a technical note or in a more detailed paper with an adequately developed description of informatic methods, data, and algorithms.
If the authors prefer the original article format, I suggest reformate it to a brief report as proof of concept.

Author Response

We thank Reviewer 3 for their feedback.

We have included in the main text (in the abstract and conclusion sections) clear links to BioTEA's code and usage guides. We hope that direct inspection of the code will give a more detailed overview of BioTEA's scope.
We were not aware that the Technical Note format was available for the journal Biology. We are open to modifing the manuscript type from Article to Technical Note if the Editor deems it necessary and viable. However, we think that our current manuscript already follows MDPI's requirements for a technical note, so we believe that minimal editing would be needed following such a change.

Round 2

Reviewer 3 Report

The authors incorporated the required information